# Age-related differences in correction behavior for unintended acceleration

**Kunihiro Hasegawa** *, **Motohiro Kimura, Yuji Takeda**

Department of Information Technology and Human Factors, National Institute of Advanced Industrial Science and Technology, Tsukuba, Ibaraki, Japan

* hasegawa.kunihiro@aist.go.jp

**Data Availability Statement:** De-identified data is posted at [https://osf.io/mre6y/?view_only=5af130ecc3234435b7a7ffc1ba6c391f]. The materials used in this study are widely available.

**Funding:** The authors received no specific funding for this work.

## Abstract

Although unintended acceleration caused by pedal misapplication is a cause of traffic accidents, fatal accidents may be avoided if drivers realize their error immediately and quickly correct how they are stepping on the pedal. This correction behavior may decline with age because the rate of fatal accidents is fairly higher for older adults than for younger adults. To investigate this possibility, the present study recruited older adults ($n$ = 40, age range = 67–81 years) as well as younger adults ($n$ = 40, age range = 18–32 years). In this study, they performed a pedal stepping task during which they were required to stop the simulated vehicle as quickly as possible when a red signal was presented on a monitor. During most trials, the vehicle decelerated/stopped when the brake pedal was applied in a normal manner. In a few trials, however, stepping on the brake pedal resulted in sudden acceleration of the vehicle (i.e., the occurrence of the unintended acceleration); when this occurred, the participants had to release the pedal and re-step on another pedal to decelerate/stop the vehicle as quickly as possible. We focused on the age-related differences of the reaction latencies during three time periods: from the appearance of the red signal on the screen until stepping on the pedal (Period 1), from stepping on the pedal until the release of the pedal (Period 2), and from the release of the pedal until re-stepping of another pedal (Period 3). The results showed that there was no age-related difference in the latency of Period 1, $p$ = .771, whereas those of Periods 2 and 3 were longer for the older adults ($p$s < .001). The results suggest that there are age-related differences in error detection and correction abilities under unintended situations with foot pedal manipulation.

## Introduction

Manipulation error is one of the major causes of serious traffic accidents; in 2016, in Japan, it accounted for 18% of all fatal traffic accidents [1]. One particular type of manipulation error—unintended acceleration due to pedal misapplication—has gained substantial public attention because many horrible accidents caused by unintended acceleration have been sensationally reported. For example, in one instance, a parked vehicle suddenly accelerated at full throttle and crashed into pedestrians, other vehicles, and buildings [2, 3].

**Competing interests:** The authors have declared that no competing interests exist.

Although dangerous, unintended acceleration itself does not always lead to serious accidents. Rather, the severity level depends on the driver's ability to stop the unintended acceleration from leading to serious accidents; it has been shown that serious accidents could have been avoided if the drivers had detected the unintended acceleration in time and promptly corrected the pedal stepping [4, 5]. Drivers who caused serious accidents continued pressing the accelerator pedal for 1–12 seconds [4]. Therefore, serious accidents caused by unintended acceleration are thought to be, at least partly, attributable to the drivers' poor abilities to detect unintended acceleration and correct the pedal stepping.

Compared to those of younger adults, the abilities of older adults to detect unintended acceleration and correct the pedal stepping may be reduced. An accident analysis performed in Japan suggested that older drivers have an inferior ability to stop unintended acceleration from resulting in serious accidents [6]. The rates of fatal and serious accidents among all traffic accidents have been reported as almost similar (6.0% vs. 5.1%, respectively) for older (older than 75 years) and younger (younger than 24 years) drivers. However, the rates of fatal and serious traffic accidents specifically caused by unintended acceleration of vehicles are fairly higher among older drivers compared to those of younger ones (5.7% and 1.7%, respectively). These data suggest that the accident becomes more serious when unintended acceleration is caused by older drivers compared to when it is caused by younger drivers. Therefore, we postulated that the abilities to detect unintended acceleration and correct pedal stepping decline with age, resulting in the reduced ability of older drivers to avoid accidents.

In addition, although not focused on pedal misapplication, many laboratory-based empirical studies have shown that the ability to detect response errors declines with age [7–12], suggesting that the ability to detect unintended acceleration during driving may also decline with age. Therefore, the aging of abilities for older adults to detect unintended acceleration may be related to the higher rate of fatal accidents in older adults. However, little is known about age-related differences in error detection and correction abilities with foot pedal manipulation.

The purpose of the present study was to determine the effects of age on correcting behavior for the unintended acceleration. First, the acceleration with respect to the intention of an operator should be simulated in the laboratory. To this end, we developed a pedal stepping task during which correcting behaviors could be evaluated. During this task, participants viewed a display simulating optic flow corresponding to driving at a constant speed on a one-lane road; they were asked to stop the optic flow by using pedal stepping as quickly as possible when a red signal was presented on the screen. In most trials (90%), the center pedal acted as a brake (a decelerating condition); by stepping on that pedal, the speed of the optic flow decreased and stopped. However, in a few trials (10%), the center pedal suddenly acted as an accelerator (an accelerating condition); as a result, the speed of the optic flow did not decrease. Instead, it increased when the participants stepped on the pedal (occurrence of unintended acceleration). When the participants became aware of the acceleration, they had to release the center pedal and re-step on the left pedal as quickly as possible (the left pedal served as the brake pedal only during the accelerating condition). Although traffic accidents could be induced not only by the own vehicle's condition but also by other vehicles, weather, and road conditions [13–15], in the present study, we used the simple simulation task to examine purely the correction behavior for unintended acceleration.

In the present study, the time course of a participant's responses in the accelerating condition was divided into three periods because the age-related decline of older adults can appear differently during different stages of processing. For example, previous psychological studies have demonstrated that stimulus detection and simple reactions (i.e., simple situations) decline less rapidly with age than choice reactions (i.e., complex situations) [16–19]. Previous

## A: decelerating condition (90% of all trials)

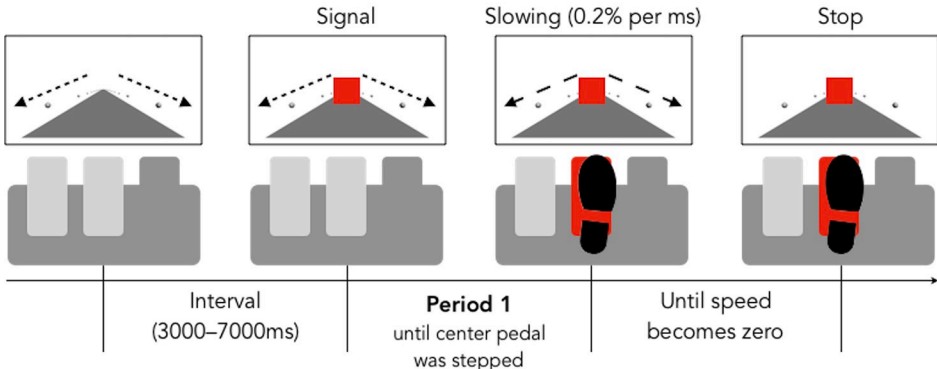

## B: accelerating condition (10% of all trials)

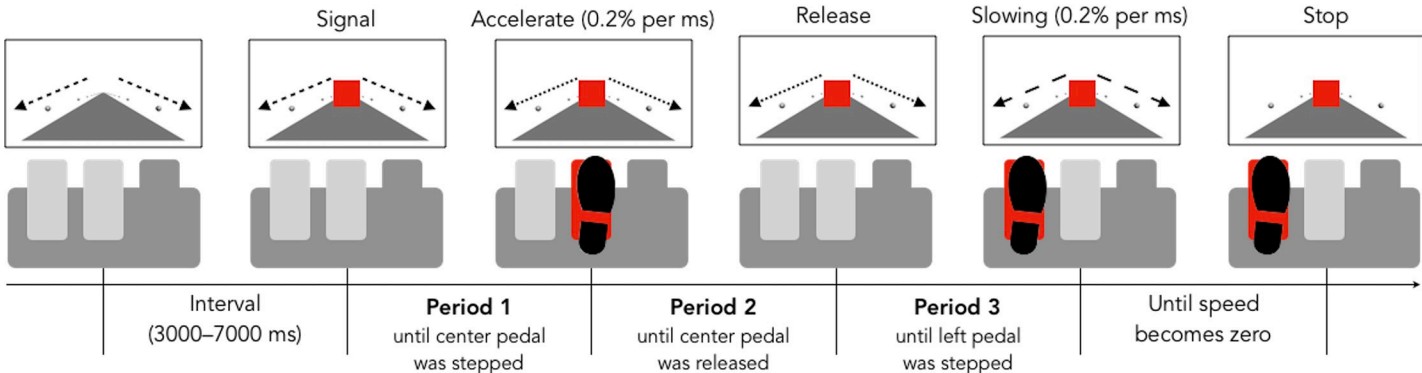

**Fig 1.** Schematic illustrations of trials for the (A) decelerating condition and (B) accelerating condition during the pedal stepping task. Participants were asked to stop the optic flow if the red signal was presented. During the decelerating condition, the speed of the optic flow decreased if the participant stepped on the center pedal. During the accelerating condition, the optic flow speed increased if the participant stepped on the center pedal; it decreased if the participant stepped on the left pedal. The two arrows on each screen were illustrated for descriptive purposes only and did not appear on the actual screen.

transportation engineering studies have also proposed similar results; there was an age-related decline in the perception-reaction time for avoidance under a complex hazard situation, while there was less age-related decline under a simple hazard situation [20–23]. In addition, the decline of a specific/non-specific cognitive function was indicated during complex tasks [24, 25]. Thus, it is important to identify the specific processes that are difficult for older adults. To approach this issue, first, we measured the latency during the period from the appearance of the red signal on the screen until stepping on the center pedal (period 1; see Fig 1), which reflected the response ability during simple situations, including the physical ability of younger and older adults. If the perceptual/physical abilities declined with age, even during simple situations, then the latency during period 1 should be longer for older adults than for younger adults. Second, the latency from stepping on the center pedal until the release of the center pedal was measured (period 2; see Fig 1). If the ability to detect the unintended acceleration and/or stop the incorrect behavior declines with age, then the latency during period 2 should be longer for older adults than for younger adults. Third, latency during the period from the release of the center pedal until re-stepping on the left pedal was estimated (period 3; see Fig 1). If the ability to execute the correct behavior declines with age, then the latency during period 3 should be longer for older adults. During this experiment, period 2 includes not only the time period including the correcting behavior (release of the center pedal) but also the

time period including the detection of the unintended acceleration; however, period 3 only reflects the time period including the correcting behavior (re-stepping on the left pedal).

Furthermore, the present study measured the subjective ratings by using several question-naires (for details, see Methods section). We analyzed correlations between these rating scores and the behavioral measures of the pedal stepping task (i.e., the latencies during periods 2 and 3). Correlation analyses were completely exploratory, and there was no specific hypothesis. If a subjective rating was significantly correlated with the behavioral measures, then it was sug-gested that the subjective rating might be useful for predicting the ability to avoid accidents when unintended acceleration occurred.

## Methods

### Ethics statement

This research complied with the Declaration of Helsinki and was approved by the Institutional Review Board at the National Institute of Advanced Industrial Science and Technology (AIST). Informed consent was obtained from each participant.

### Open practices statement

De-identified data is posted at [https://osf.io/mre6y/?view_only= 5af130ecc3234435b7a7ffc1ba6c391f]. The materials used in this study are widely available.

### Participants

Forty younger adults (18 females and 22 males; mean age = 21.90 years and age-range = 18–32 years) and 40 older adults (20 females and 20 males; mean age = 71.35 years and age-range = 67–81 years) participated in this experiment. All participants had normal vision or vision that was corrected to normal; they were not aware of the purpose of the experiment. All participants had a valid driving license and drove more than 3 days per week during the latest 6 months. Participants underwent two neuropsychological tests before the experiment: The Mini-Mental State Examination [26] and the Clock Drawing Test [27]. The results of these tests are shown in Table 1. We had been planned that the participants in which they scored under 24 points on the Mini-Mental State Examination were excluded from the data analyses. However, all participants scored over 24 points on the Mini-Mental State Examination and received full marks in the clock drawing test, suggesting that older, as well as younger partici-pants, were neuropsychologically healthy. Therefore, any participant was not excluded from the data analyses.

### Apparatus and stimuli

The pedal stepping task was controlled by a laptop computer (MacBook Pro; Apple) using MATLAB (MathWorks) and Psychophysics Toolbox 3 [28, 29] including OpenGL (Khronos

**Table 1. Results of neuropsychological tests.**

| Neuropsychological test | Younger (n = 40) | | Older (n = 40) | |
|---|---|---|---|---|
| | *Mean* | *SD* | *Mean* | *SD* |
| Mini-Mental State Examination | 29.82 | 0.38 | 29.30 | 1.09 |
| Clock Drawing Test [a] | 10.00 | — | 10.00 | — |

[a]All participants achieved perfect scores on the Clock Drawing Test.

Group). The visual stimuli (Fig 1) were presented on a screen with a 23.6-inch liquid crystal display (XL2410T; BenQ) with a viewing distance of approximately 100 cm. The pedal stepping responses were recorded by the pedal box of the racing game controller (Driving Force G29; Logitech) with the "gamepad" command of Psychophysics Toolbox 3.

A simple road scene (Fig 1) was drawn on the screen by using OpenGL. The size and arrangement of all objects were calculated by the functions of the perspective projection mode in OpenGL. The road was drawn toward the road dissipation point (the center of the screen) from the lower right, and lower left corners of the screen. Additionally, the spheres were drawn on each side of the road and arranged at equal intervals in terms of the three-dimensional expression. The background was drawn in white, and the road and spheres were drawn in neutral gray.

## Procedure

The pedal stepping task consisted of 400 trials: 360 trials involving the decelerating condition during which the center pedal acted as a brake and 40 trials involving the accelerating condition during which the center pedal acted as an accelerator. The 400 trials were performed in random order. Participants could rest after every 40 trials. Before the start of the task, participants were informed about the acceleration and deceleration functions of the pedals and were given instructions regarding how to perform pedal stepping during each condition. After the experiment, we also measured subjective factors using several questionnaires as an exploratory investigation. However, these were excluded from the article because we concerned that it deviated from the core principle of this article and lacked the statistical power in the analyses. The details were shown in the S1 File.

Schematic illustrations of the trials during the decelerating and accelerating conditions are shown in Fig 1. Trials involving both the decelerating and accelerating conditions started with the presentation of the optic flow at a constant speed; the optic flow was simulated to be the same as what happens when the vehicle runs at approximately 30 kilometers per hour. Participants were required to put their right foot on the center pedal of the pedal box and wait for the presentation of a red signal (a red square) on the screen. Following a random interval of 3,000–7,000 milliseconds, the red signal was presented at the center of the screen, and the participants were required to step on the center pedal as quickly as possible.

During the trials involving the decelerating condition (90% of trials), the center pedal acted as a brake. Therefore, if the center pedal was stepped on, then the speed of the optic flow decreased by 0.2% of the initial speed per millisecond. Participants were required to keep stepping on the brake pedal until the optic flow speed became zero; it required approximately 500 milliseconds because the participants started to step on the center pedal.

During the trials involving the accelerating condition (10% of trials), the center pedal acted as an accelerator. Therefore, if the center pedal was stepped on, the speed of the optic flow increased by 0.2% of the initial speed per millisecond. The increment of the optic flow speed was a cue to be aware that the center pedal had acted as an accelerator. If the participants were aware of the increment in the flow speed, then they were required to release the center pedal and to re-step on the left pedal of the pedal box as quickly as possible.

## Statistical analysis

Mean latencies were analyzed by a mixed two-way analysis of variance (ANOVA) with age (younger and older; between-subject factor) and period (1, 2, and 3; within-subject factor) as the factors. A post hoc analysis was performed using an independent sample $t$-test with age (younger and older) as the factor. To directly compare the magnitudes of age-related effects

during periods 2 and 3, we performed a mixed two-way ANOVA with age (younger and older; between-subject factor) and period (2 and 3; within-subject factor) as factors.

## Results

### Decelerating condition

The pedal stepping performance during the deceleration condition was shown in Fig 2A. This was analyzed by an independent sample $t$-test with age (younger and older). As a result, there was no significant difference between the younger and older adults, $t(78) = 0.07$, $p = .943$, $d = -0.02$, 95% CI = [−78.8, 73.3].

### Accelerating condition

The pedal stepping performance during the accelerating condition was shown in Fig 2B. These were analyzed in terms of the mean latencies during the three periods described in the Methods section. The main effect of age was significant, $F(1, 78) = 41.69$, $p < .001$, $\omega^2 = 0.34$. The main effect of period was significant, $F(2, 156) = 168.98$, $p < .001$, $\omega^2 = 0.46$. The interaction between age and period was also significant, $F(2, 156) = 44.41$, $p < .001$, $\omega^2 = 0.18$. A post hoc analysis using independent sample $t$-tests with age (younger and older) as the factor indicated that the mean latencies during period 1 did not significantly differ between the younger and older adults, $t(78) = -0.29$, $p = .771$, $d = -0.07$, 95% CI = [−85.6, 63.7], whereas those of period 2 were longer for the older adults, $t(78) = 7.08$, $p < .001$, $d = 1.58$, 95% CI = [209.4, 373.2]; and those of period 3 were also longer for older adults, $t(78) = 7.08$, $p < .001$, $d = 1.58$, 95%

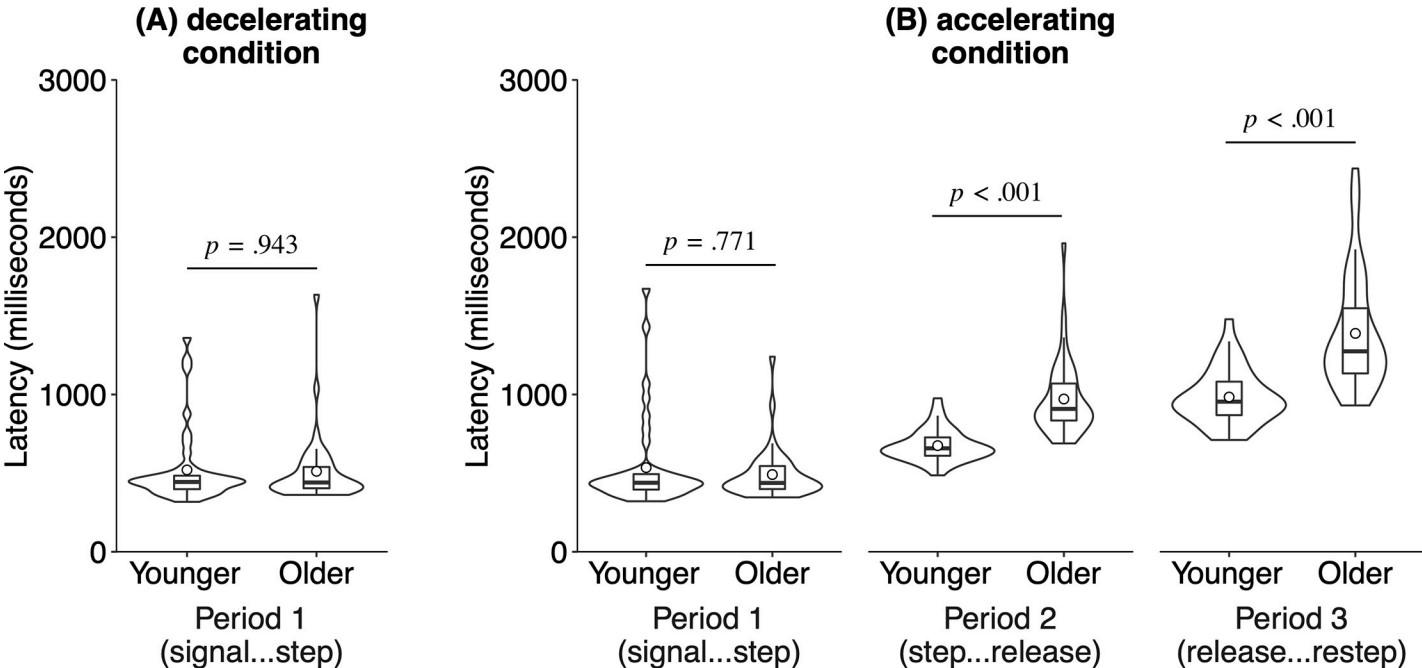

**Fig 2. Results of the pedal stepping tasks.** (A) Results of the decelerating condition. (B) Results of the acceleration condition. Period 1: the period from the appearance of the red signal on the screen until stepping on the center pedal. Period 2: the period from stepping on the center pedal until the release of the center pedal. Period 3: the period from the release of the center pedal until re-stepping on the left pedal. The white dot indicates the mean of latency in each condition. The bold horizontal line in the boxplot indicates the median latency in each condition. The box in the boxplot indicates the lower (Q1) and upper (Q3) quartile range (IQR; Inter-Quartile Range). The whisker in the boxplot indicates the lower (Q1 − 1.5 × IQR) and upper (Q3 + 1.5 × IQR) extreme ranges. The violin plot indicates the distribution of the mean latency of each participant in each period and age group.

CI = [407.6, 726.2]. To directly compare the magnitudes of age-related effects during periods 2 and 3, we performed a mixed two-way ANOVA with age (younger and older; between-subject factor) and period (2 and 3; within-subject factor) as factors. The results showed a significant interaction, $F(1, 78) = 20.13$, $p < .001$, $\omega^2 = 0.05$, indicating that the age-related difference of period 3 was significantly larger than that of period 2.

## Discussion

The present study showed that the latencies during period 2 (i.e., the period from stepping on the center pedal until release of the center pedal) and period 3 (the period from release of the center pedal until re-stepping on the left pedal) were longer for older adults compared to those of younger adults; however, there was no significant difference in the latencies during Period 1 (from the appearance of the red signal on the screen until stepping on the center pedal) between the younger and older adults. The absence of any significant differences in latencies during period 1 suggests that the visual perception (detecting the red signal) and simple physical reaction (stepping on the pedal) of older adults were relatively intact compared with those of younger adults. As mentioned, previous psychological studies, as well as transportation engineering studies, showed that there is a little age-related decline in simple reaction (i.e., simple situations) [16–23]. The present finding—the absence of an age-related difference in latencies during period 1—is mostly consistent with the findings of these previous studies.

A significant difference in the latencies during period 2 indicates that the ability to detect unintended acceleration and/or stop the incorrect behavior declined with age. Although not related to a driving situation, previous empirical studies demonstrated an age-related decline in the detection of response error [7–12]. Therefore, the present finding during period 2 is consistent with that of previous reports. Furthermore, a significant difference in latencies during period 3 indicates that the ability to execute the re-stepping task declined with age. Interestingly, the age-related decline was greater for the time period involving re-stepping on the left pedal (period 3) than for the time period between detection of the unintended acceleration and release of the center pedal (period 2). The mean latency for older adults during period 3 was 566.9 ms longer than that of the younger adults, which corresponds to two-thirds of the total meantime for age-related slowing (291.3 ms during period 2 + 566.9 ms during period 3 = 858.2 ms total).

One of the possible causes of prolonged latencies of older adults during period 2 and/or period 3 could be the age-related decline in their mental ability to switch from the regular stepping task (stepping on the center pedal) to the re-stepping task because it is well known that the ability to manage multiple tasks considerably declines with age [30]. Regarding period 3, it is possible that choosing the correct pedal would have been difficult for older adults. During period 1, stepping on the center pedal was the only choice of reaction for the participants because they put their right foot on the center pedal while they were waiting for a red signal. However, during period 3, participants had to choose the correct pedal (the left pedal) from among three pedals of the pedal box while their right foot was free after the pedal was released during Period 2. In previous studies, the inhibitory deficit hypothesis (the decline of a specific cognitive function) [24] and the general slowing hypothesis (the decline of non-specific cognitive function) [25] were discussed as the factors associated with the age-related decrease in the ability to perform the task of making a complex choice. To choose the correct pedal, the inhibitory function is needed to identify and exclude the incorrect choice options or general slowing of deciding on choice will occur. It is difficult to dissociate these effects in the present experiment; however, either factor may have affected the latencies during period 3. Additionally, it is possible that the physical ability to shift the foot from the center pedal to the left pedal declined with age. Although further studies are needed to clarify the cause of the age-related slowing

during period 3, the present findings may provide an important clue to reducing serious accidents caused by older adults during unintended acceleration events. The magnitude of the age-related decline of latency during period 2 (latency until release of the incorrect pedal) was relatively low; therefore, it may be possible to prevent some of serious accidents if quick deceleration occurs after releasing the accelerator pedal.

One important aspect of the present findings is that the age-related decline was observed in neurologically healthy older adults (Table 1). Recently, in Japan, to prevent traffic accidents, older adults (older than 75 years) are required to complete a cognitive assessment test (a type of dementia test) when they renew their license to drive. However, approximately half of the older adults who caused fatal traffic accidents in 2018 in Japan had shown no signs of decline in their cognitive ability according to the cognitive assessment test [31], indicating that the current cognitive assessment test might not precisely assess the accident risks. The current findings appear to be consistent with this previous study because there is a clear sign of age-related decline in the mental ability of neuropsychologically healthy older adults who attempted to perform the regular stepping task. It remains to be verified whether the behavioral performance during the pedal task used in this study can precisely reflect drivers' behaviors during unintended acceleration situations in the real world. It also remains to be determined to what extent the prolongation of latencies during periods 2 and 3 is related to the severity of traffic accidents caused by unintended acceleration. Other than the age-related decline examined in the present study, there may be some other factors, such as steering to avoid the crash, that account for older drivers' reduced ability to avoid accidents during unintended acceleration events. To gain a better understanding of the accident risks during unintended acceleration situations, these issues need to be addressed in future studies.

Finally, it should be noted that the change in the optical flow speed at the center of the visual field was the only cue for the detection of unintended acceleration during the present study. In the real world, drivers may detect the unintended acceleration using other cues (e.g., loud engine sound, violent response of the speedometer, and others). However, a previous study demonstrated that the sensitivity of the radial optical flow is not prone to age-related decline [32]. This suggests that the differences in the correcting ability during unintended acceleration for adults are based on a more later cognitive/behavioral factors rather than early sensational/perceptual factors. Therefore, we consider that the present findings would not be limited by the modality of the cue for the detection of unintended acceleration.

## Conclusion

The present study demonstrated that during unintended acceleration situations, the ability to correct pedal stepping declined in older; however, no significant age-related decline was found for the quickness of performing regular and simple pedal stepping. Furthermore, we examined the age-related differences in the time period until the release of the wrong pedal and re-stepping on the correct pedal separately; we found that, compared with the former, the latter was considerably affected by age.

## Supporting information

**S1 File.**
(DOCX)

## Author Contributions

**Conceptualization:** Kunihiro Hasegawa, Motohiro Kimura, Yuji Takeda.

**Formal analysis:** Kunihiro Hasegawa.

**Investigation:** Kunihiro Hasegawa.

**Methodology:** Kunihiro Hasegawa.

**Project administration:** Kunihiro Hasegawa.

**Supervision:** Motohiro Kimura, Yuji Takeda.

**Validation:** Kunihiro Hasegawa.

**Visualization:** Kunihiro Hasegawa.

**Writing – original draft:** Kunihiro Hasegawa.

**Writing – review & editing:** Motohiro Kimura, Yuji Takeda.

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
