## [Decision Letter · Decision Letter 0]

16 Apr 2020

PONE-D-20-07881

Age-related differences in correction behavior for unintended acceleration

PLOS ONE

Dear Dr Hasegawa,

Thank you for submitting your manuscript to PLOS ONE. After careful consideration, we feel that it has merit but does not fully meet PLOS ONE’s publication criteria as it currently stands. Therefore, we invite you to submit a revised version of the manuscript that addresses the points raised during the review process.

We would appreciate receiving your revised manuscript by May 31 2020 11:59PM. To enhance the reproducibility of your results, we recommend that if applicable you deposit your laboratory protocols in protocols.io, where a protocol can be assigned its own identifier (DOI) such that it can be cited independently in the future. For instructions see: http://journals.plos.org/plosone/s/submission-guidelines#loc-laboratory-protocols

We look forward to receiving your revised manuscript.

Kind regards,

Feng Chen

Academic Editor

PLOS ONE

2. Please provide additional details regarding participant consent. In the ethics statement in the Methods and online submission information, please ensure that you have specified what type of consent you obtained (for instance, written or verbal, and if verbal, how it was documented and witnessed).

Reviewers' comments:

Reviewer's Responses to Questions

**Comments to the Author**

1. Is the manuscript technically sound, and do the data support the conclusions?

Reviewer #1: Partly

Reviewer #2: Partly

2. Has the statistical analysis been performed appropriately and rigorously? 

Reviewer #1: N/A

Reviewer #2: Yes

3. Have the authors made all data underlying the findings in their manuscript fully available?

Reviewer #1: Yes

Reviewer #2: Yes

4. Is the manuscript presented in an intelligible fashion and written in standard English?

Reviewer #1: Yes

Reviewer #2: Yes

5. Review Comments to the Author

Reviewer #1: It's an interesting topic, here are some suggestions for the authors to consider.

1.The paper structure is poor. It should be reorganized carefully, such as summarizing the research purpose in brief in the introduction section, adding conclusion section, rewriting the abstract, summarizing the testing scheme in a table, etc.

2.Line 227-234, the conclusion is doubtful. There are also lots of previous studies proved that the perception-reaction time for the elderly is greater than the younger adults, especially in the transportation engineering field. Please providing more proofs and discussion on this point.

3.In the Apparatus and Stimuli section, present the apparatus and testing scene in figures. The study used a racing game controller to conduct the test, maybe it’s economical but a little bit simple. Are there any previous studies using such similar scheme? How to prove the reliability?

4.Table 2, illustrating the data in some figures to provide more information, not only the Mean and SD.

Reviewer #2: The manuscript investigated the research question whether aging can impact the correction behavior for unintended accelerations. This study is properly organized. There are several concerns with the paper as listed below:

1. The abstract is not properly written. It should be concise and comprehensive. The main body of the abstract doesn't even include results from the statistical analysis:"During most trials, the vehicle decelerated/stopped when the brake pedal was applied in a normal manner. In a few trials, however, stepping on the brake pedal resulted in sudden acceleration of the vehicle (i.e., the occurrence of the unintended acceleration); when this occurred, the participants had to release the pedal and re-step on another pedal to decelerate/stop the vehicle as quickly as possible. The latency between the onset of the unintended acceleration and the release of the pedal and the period from the release of the pedal until re-stepping on another pedal were found to be longer for older adults. "

2. The conclusion that "This suggests that age-related declines in the correction behavior may be the main cause of the fatal accidents caused by unintended acceleration. " is not supported by the study.

3. The literature review is not exhaustive, several papers should be acknowledged and cited in the paper:

[1] Feng Chen, Haorong Peng, Xiaoxiang Ma, Jieyu Liang, Wei Hao, Xiaodong Pan.“Examining the safety of trucks under crosswind at bridge-tunnel section: A driving simulator study”, Tunnelling and Underground Space Technology, 2019, 92, 103034. https://doi.org/10.1016/j.tust.2019.103034

[2] Chen, Feng, Mingtao Song, Xiaoxiang Ma, and Xingyi Zhu. “Assess the Impacts of Different Autonomous Trucks’ Lateral Control Modes on Asphalt Pavement Performance.” Transportation Research C: Emerging Technologies，2019, 103, 17-29.

[3] Bowen Dong, Xiaoxiang Ma, Feng Chen and Suren Chen. “Investigating the Differences of Single- and Multi-vehicle Accident Probability Using Mixed Logit Model", Journal of Advanced Transportation, 2018, UNSP 2702360.

6. PLOS authors have the option to publish the peer review history of their article (what does this mean?). If published, this will include your full peer review and any attached files.

Reviewer #1: No

Reviewer #2: No

---

## [Author Response · Author response to Decision Letter 0]

5 Jun 2020

Paper structure

[Reviewer 1] 

The paper structure is poor. It should be reorganized carefully, such as summarizing the research purpose in brief in the introduction section, adding conclusion section, rewriting the abstract, summarizing the testing scheme in a table, etc. 

[Reviewer 2] 

The abstract is not properly written. It should be concise and comprehensive. The main body of the abstract doesn't even include results from the statistical analysis: "During most trials, the vehicle decelerated/stopped when the brake pedal was applied in a normal manner. In a few trials, however, stepping on the brake pedal resulted in sudden acceleration of the vehicle (i.e., the occurrence of the unintended acceleration); when this occurred, the participants had to release the pedal and re-step on another pedal to decelerate/stop the vehicle as quickly as possible. The latency between the onset of the unintended acceleration and the release of the pedal and the period from the release of the pedal until re-stepping on another pedal were found to be longer for older adults."

[Response]

We revised the abstract, including the detailed results (lines 15–22 in the revised version of the manuscripts). We also added the brief purpose of the present study in the Introduction (line 60 in the revised version of the manuscripts). Furthermore, the Conclusion section was added to the end of the Discussion section (line 324 in the revised version of the manuscripts). 

Unfortunately, we could not recognize the meaning of “summarizing the testing scheme in a table.” Please let me know the meaning of “summarizing the testing scheme in a table,” if it is needed.

About the simple reaction times in older adults

[Reviewer 1] 

Line 227-234, the conclusion is doubtful. There are also lots of previous studies proved that the perception-reaction time for the elderly is greater than the younger adults, especially in the transportation engineering field. Please providing more proofs and discussion on this point.

[Response]

We agree that some previous studies in transportation engineering showed the age-difference of the perception-reaction time (e.g., Salvia et al., 2016; Svetina, 2016). However, it was also shown that the age-related effect in perception-reaction times varied depending on task complexity (Lerner, 1993); there was no age-difference of the perception-reaction time in a simple hazard situation (Olson & Sivak, 1986; Salvia et al., 2016). This dependency on the task complexity is consistent with the evidence in psychological studies, which are cited in the present paper. We added the mentions of this issue (see lines 83–86 and 251–253 in the revised version of the manuscript).

Ref. 

Lerner, N. D. (1993). Brake perception-reaction times of older and younger drivers. Proc. Hum. Factors Ergon. Soc. Annu. Meet., 37, 206–210. doi:10.1177/154193129303700211

Olson, P. L., & Sivak, M. (1986). Perception-response time to unexpected roadway hazards. Human Factors, 28, 91–96. doi:10.1177/001872088602800110

Salvia, E., Petit, C., Champely, S., Chomette, R., Di Rienzo, F. & Collet, C. (2016). Effects of age and task load on drivers’ response accuracy and reaction time when responding to traffic lights. Frontiers in Aging Neuroscience, 8, 169. doi:10.3389/fnagi.2016.00169

Svetina, M. (2016). The reaction times of drivers aged 20 to 80 during a divided attention driving, Traffic Injury Prevention, 17, 810–814, doi:10.1080/15389588.2016.1157590

The reliability to use a game controller

[Reviewer 1] 

In the Apparatus and Stimuli section, present the apparatus and testing scene in figures. The study used a racing game controller to conduct the test, maybe it’s economical but a little bit simple. Are there any previous studies using such similar scheme? How to prove the reliability?

[Response]

A game controller is a common tool for reaction time study in experimental psychology, and that is officially supported by the “gamepad” command implemented in Psychophysics Toolbox 3 (Brainard, 1997). We added this point in lines 153–155 in the revised version of the manuscript. A racing game controller is also used in a study of driving ability (e.g., Mackenzie & Harris, 2017; Salvia et al., 2016). We believe a racing game controller was sufficient because no detailed operation was required, just required "step" or "release" of the pedals in the present study.

Ref. 

Brainard, D. H. (1997). The Psychophysics Toolbox. Spatial Vision, 10(4), 433–436. https://doi.org/10.1163/156856897X00357

Mackenzie, A. K., & Harris, J. M. (2017). A link between attentional function, effective eye movements, and driving ability. Journal of Experimental Psychology: Human Perception and Performance, 43(2), 381–394. http://dx.doi.org/10.1037/xhp0000297

Salvia, E., Petit, C., Champely, S., Chomette, R., Di Rienzo, F. & Collet, C. (2016). Effects of age and task load on drivers’ response accuracy and reaction time when responding to traffic lights. Frontiers in Aging Neuroscience, 8, 169. doi:10.3389/fnagi.2016.00169

Data visualization

[Reviewer 1] 

Table 2, illustrating the data in some figures to provide more information, not only the Mean and SD. 

We visualized the data (see Fig 2 of the revised version of the manuscript). 

Conclusion of the present study

[Reviewer 2] 

The conclusion that "This suggests that age-related declines in the correction behavior may be the main cause of the fatal accidents caused by unintended acceleration. " is not supported by the study.

[Response]

We deleted this mention in the conclusion section. In addition, we relaxed our claim regarding the relationship between the present study and the traffic accidents in the Introduction sections (lines 56–59 in the revised version of the manuscripts). 

The literature reviews

[Reviewer 2] 

The literature review is not exhaustive, several papers should be acknowledged and cited in the paper: [1] Feng Chen, Haorong Peng, Xiaoxiang Ma, Jieyu Liang, Wei Hao, Xiaodong Pan.“Examining the safety of trucks under crosswind at bridge-tunnel section: A driving simulator study”, Tunnelling and Underground Space Technology, 2019, 92, 103034. https://doi.org/10.1016/j.tust.2019.103034 [2] Chen, Feng, Mingtao Song, Xiaoxiang Ma, and Xingyi Zhu. “Assess the Impacts of Different Autonomous Trucks’ Lateral Control Modes on Asphalt Pavement Performance.” Transportation Research C: Emerging Technologies，2019, 103, 17-29. [3] Bowen Dong, Xiaoxiang Ma, Feng Chen and Suren Chen. “Investigating the Differences of Single- and Multi-vehicle Accident Probability Using Mixed Logit Model", Journal of Advanced Transportation, 2018, UNSP 2702360.

[Response]

We added your recommended articles (see lines 74–77 in the revised version of the manuscript).

---

## [Decision Letter · Decision Letter 1]

29 Jun 2020

Age-related differences in correction behavior for unintended acceleration

PONE-D-20-07881R1

Dear Dr. Hasegawa,

We’re pleased to inform you that your manuscript has been judged scientifically suitable for publication and will be formally accepted for publication once it meets all outstanding technical requirements.

Kind regards,

Feng Chen

Academic Editor

PLOS ONE

Additional Editor Comments (optional):

Reviewers' comments:

Reviewer's Responses to Questions

**Comments to the Author**

1. If the authors have adequately addressed your comments raised in a previous round of review and you feel that this manuscript is now acceptable for publication, you may indicate that here to bypass the “Comments to the Author” section, enter your conflict of interest statement in the “Confidential to Editor” section, and submit your "Accept" recommendation.

Reviewer #1: All comments have been addressed

Reviewer #2: All comments have been addressed

2. Is the manuscript technically sound, and do the data support the conclusions?

Reviewer #1: Yes

Reviewer #2: Yes

3. Has the statistical analysis been performed appropriately and rigorously? 

Reviewer #1: Yes

Reviewer #2: Yes

4. Have the authors made all data underlying the findings in their manuscript fully available?

Reviewer #1: Yes

Reviewer #2: Yes

5. Is the manuscript presented in an intelligible fashion and written in standard English?

Reviewer #1: Yes

Reviewer #2: Yes

6. Review Comments to the Author

Reviewer #1: (No Response)

Reviewer #2: (No Response)

7. PLOS authors have the option to publish the peer review history of their article (what does this mean?). If published, this will include your full peer review and any attached files.

Reviewer #1: No

Reviewer #2: No

---

## [Editor Report · Acceptance letter]

1 Jul 2020

PONE-D-20-07881R1 

Age-related differences in correction behavior for unintended acceleration 

Dear Dr. Hasegawa:

I'm pleased to inform you that your manuscript has been deemed suitable for publication in PLOS ONE. Congratulations! Your manuscript is now with our production department. 

Kind regards, 

on behalf of

Dr. Feng Chen 

Academic Editor

PLOS ONE